# In Vitro Characterization and Antiviral Susceptibility of Ophidian Serpentoviruses

**DOI:** 10.3390/microorganisms11061371

**Published:** 2023-05-24

**Authors:** Steven B. Tillis, Camille Holt, Spencer Havens, Tracey D. Logan, Justin G. Julander, Robert J. Ossiboff

**Affiliations:** 1Department of Comparative, Diagnostic, and Population Medicine, College of Veterinary Medicine, University of Florida, Gainesville, FL 32608, USA; reptillis@ufl.edu (S.B.T.); tracey.logan@phhp.ufl.edu (T.D.L.); 2Animal, Dairy, and Veterinary Sciences, College of Agriculture and Applied Sciences, Utah State University, Logan, UT 84322, USA; holt.camille@gmail.com (C.H.); spencer.havens@usu.edu (S.H.); justin.julander@usu.edu (J.G.J.)

**Keywords:** antiviral, growth kinetics, nidovirus, Python, reptile, RNA virus, snake

## Abstract

Ophidian serpentoviruses, positive-sense RNA viruses in the order *Nidovirales*, are important infectious agents of both captive and free-ranging reptiles. Although the clinical significance of these viruses can be variable, some serpentoviruses are pathogenic and potentially fatal in captive snakes. While serpentoviral diversity and disease potential are well documented, little is known about the fundamental properties of these viruses, including their potential host ranges, kinetics of growth, environmental stability, and susceptibility to common disinfectants and viricides. To address this, three serpentoviruses were isolated in culture from three unique PCR-positive python species: Ball python (*Python regius*), green tree python (*Morelia viridis*), and Stimson’s python (*Antaresia stimsoni*). A median tissue culture infectious dose (TCID_50_) was established to characterize viral stability, growth, and susceptibility. All isolates showed an environmental stability of 10–12 days at room temperature (20 °C). While all three viruses produced variable peak titers on three different cell lines when incubated at 32 °C, none of the viruses detectably replicated at 35 °C. All viruses demonstrated a wide susceptibility to sanitizers, with 10% bleach, 2% chlorhexidine, and 70% ethanol inactivating the virus in one minute and 7% peroxide and a quaternary ammonium solution within three minutes. Of seven tested antiviral agents, remdesivir, ribavirin, and NITD-008, showed potent antiviral activity against the three viruses. Finally, the three isolates successfully infected 32 unique tissue culture cell lines representing different diverse reptile taxa and select mammals and birds as detected by epifluorescent immunostaining. This study represents the first characterization of in vitro properties of growth, stability, host range, and inactivation for a serpentovirus. The reported results provide the basis for procedures to mitigate the spread of serpentoviruses in captive snake colonies as well as identify potential non-pharmacologic and pharmacologic treatment options for ophidian serpentoviral infections.

## 1. Introduction

Viral pathogens of reptiles have been historically understudied and underrepresented in research [1]. However, in recent years, increased access to more affordable metagenomic sequencing methods has resulted in something of a genetic revolution in virology, allowing for the identification of a multitude of previously unknown viruses, including those of reptiles [2]. Characterization of reptile viruses is often limited to genome sequencing. Occasionally, for viruses associated with clinical disease, co-localization of viral protein or nucleic acid to affected tissues is also performed. However, more detailed reptile viral characterization investigations are infrequent. Such investigations can be limited by scarcity of available reagents, particularly tissue culture cell lines. As only a few reptile cell lines are available commercially, isolation and in vitro characterization of reptile viruses can be challenging. Nevertheless, in vitro laboratory characterization of reptile viruses has been sporadically successful, including the determination of fer-de-lance paramyxovirus (ferlavirus) in vitro host range and ethanol susceptibility as far back as 1979 [3]. More recent examples include experiments with isolates of Hartmaniviruses and other Reptarenaviruses frequently seen in captive boa and python species [4,5].

Respiratory disease is a common ailment and major cause of morbidity and mortality in captive snakes [6,7]. Viruses in the subfamily *Serpentovirinae*, single-stranded, positive-sense RNA viruses in the order *Nidovirales*, are documented to be a contributor to respiratory disease in both captive and wild reptiles [8,9,10,11]. While the host range and pathogenicity of specific serpentoviruses require additional study, on the whole, subfamily *Serpentovirinae* exhibits approximately 30 kilobase genomes with wide genetic variation and a diverse host range, with multiple genera of viruses [12] documented in turtle [9], lizard [10,13] colubrid, boa and python species [11,14,15].

The greatest serpentoviral diversity is documented in captive python species [11]. The high prevalence of these viruses, as high as 26.4% [15] to 37.7% [11] across pythons, is generally representative of a mix of clinical and subclinical infections with evidence for variable and seasonal shedding [11,14,15]. While the presence of these viruses in captive snakes may be an extension of natural pathogen-host dynamics [9,10,14], more virulent strains observed in captivity can represent a significant threat to animal health and economic investment/revenue generation [10,11,16]. While awareness regarding the existence and pathogenic potential of these viruses is widespread among those working with susceptible snakes, there is little serpentovirus-specific information available to stakeholders as it relates to viral mitigation and therapeutics. Current best practices are based on research on distantly related nidoviruses, such as coronaviruses [17,18,19,20,21]. While enveloped RNA viruses generally have limited environmental stability [17,19] and easily disrupted lipid envelopes [22], current recommendations based on surrogate viruses could be inappropriate for serpentoviruses. 

Keepers generally only have the option of permanent quarantine or euthanasia for animals that become infected with a transmissible and virulent viral pathogen. While numerous antibacterial, antifungal, and antiparasitic treatments have been identified and evaluated in reptiles, only a few antivirals have been tested to date. Acyclovir and ganciclovir were shown to be active at doses of 25 or 50 µg/mL in turtle heart cell cultures infected with a pathogenic herpesvirus that was isolated from live and dead tortoises during an outbreak [23]. A recent study has demonstrated activity of various type I interferons from a Chinese soft-shelled turtle (*Pelodiscus sinensis*) administered endogenously to soft-shelled turtle artery cells infected with soft-shelled turtle iridovirus (STIV) in culture [24].

The aim of this study was to establish in vitro assays to characterize the stability (environmental, freeze-thaw), growth kinetics (single-cycle, multi-cycle, and temperature-dependent) host range, and susceptibility (sanitizers, viricides) for three unique python serpentoviruses. Such data will help to better inform veterinarians, curators, biologists, pet owners, breeders, and/or other stakeholders when dealing with instances of serpentoviral infection in susceptible reptile populations.

## 2. Materials and Methods

### 2.1. Establishing Reptile Cell Lines

Novel reptile cell lines used in this study were generated as part of a funded project to increase the availability of reagents to be used in reptile disease research. Detailed descriptions of the creation and establishment of the cell lines will be published independently. Briefly, various tissues were collected opportunistically during postmortem examinations of freshly deceased reptiles at the University of Florida’s College of Veterinary Medicine. No animals were euthanized for the sole purpose of tissue collection. Tissue samples were minced finely and mixed with completed media to seed either 6-well plates or small tissue culture flasks. Completed growth medium consisted of minimum essential medium with Earle’s balanced salts (GenClone, San Diego, CA, USA), L-Glutamine (MEM/EBSS; GenClone), 10% heat inactivated fetal bovine serum (FBS; GenClone, San Diego, CA, USA), nonessential amino acids (Caisson, Smithfield, UT, USA), penicillin-streptomycin solution (GenClone), amphotericin B (HyClone, Logan, UT, USA), and gentamicin (GenClone).

All cell lines were maintained at 32 °C in a humidified, 5% CO_2_ atmosphere. Proliferating primary cells forming monolayers radiating from minced tissue pieces were nurtured until near-complete confluency and then passed with 0.25% trypsin (GenClone) into an appropriately sized tissue culture flask. After 12–15 passages of consistent growth, cells were considered a cell line suitable for experimentation. 

### 2.2. Serpentovirus Isolation 

All virus preparations used in this study were grown on Diamond python heart cells (DPHt). The establishment of this cell line is described in Hoon-Hanks et al. (2018). Successful isolation of serpentovirus in culture was confirmed using a touchdown rtPCR protocol and primers reported in Hoon-Hanks et al. (2019). First, RNA was extracted from cell culture lysate using the Direct-zol RNA Miniprep kit (Zymo Research, Irvine, CA, USA). The rtPCR included sense primer BarniPVTF (5′-GAG GAC TCC ACA ARC CAG TCA C-3′) and antisense primer BarniGGTR (5′-KGC ATC RCC RCT ACT TGT GCC CTC C-3′). The rtPCR mix for each primer pair included 4 µL of 10 µM forward/sense primer, 4 µL of 10 µM reverse/antisense primer, 25 µL of 2× PCRBio rt-PCR mix, 11.5 µL of H_2_O, 2.5 µL of 20Xrtase Taq, and 3 µL RNA extract. Samples were run in a MJ Research PTC-100 Thermal Cycler with conditions for each as follows: 50 °C for 10 min; 94 °C for 2 min; 94 °C for 30 s, 55 °C for 30 s and dropping 1 °C every cycle, and 72 °C for 30 s for 5 cycles; 94 °C for 30 s, 50 °C for 30 s, and 72 °C for 30 s for 40 cycles; and 72 °C for 7 min followed by holding at 4 °C. 

Products of rtPCR were visualized on a 1% agarose gel. Bands of approximately 330 base pairs in length were excised and nucleic acids were extracted using Zymo Clean Gel DNA Recovery Kit (Zymo Research, Irvine, CA, USA) per the manufacturer’s recommendation. Samples were submitted for bidirectional Sanger sequencing to a commercial facility (Genewiz, South Plainfield, NJ, USA). Sequences were edited and aligned using Geneious Prime (Auckland, New Zealand) and considered positive if a serpentovirus sequence was returned as the closest match on NCBI BLASTN. 

### 2.3. Ball Python Nidovirus

The isolation of Ball Python Nidovirus 1 (BPNV), from host ball python (*Python regius*) is described in Hoon-Hanks et al. (2018). BPNV inoculum used for characterization in this study was unfiltered lysate of a passage 4 preparation. DPHt cells inoculated with BPNV lysate were frozen at −80 °C when cells in the flasks showed peak cytopathic effect (CPE) approximately four to five days post-inoculation. Genetic sequence of this isolate can be found at Genbank, accession number KJ541759. 

### 2.4. Green Tree Python Serpentovirus

The green tree python serpentovirus (GTSV) isolate in this study was isolated from a diseased green tree python (*Morelia viridis*). Growth of the virus was confirmed using the modified rtPCR PVTF/GGTR protocol described above in the lysate of multiple passages. GTSV inoculum used for characterization was unfiltered lysate of a passage 4 preparation. DPHt cells inoculated with GTSV lysate were frozen when cells in the flasks showed peak CPE at approximately four to five days post-inoculation. Sanger sequencing placed the closest match to this isolate as a green tree python serpentovirus with Genbank accession number MN161569.

### 2.5. Antaresian Python Serpentovirus

The antaresian python serpentovirus (AnPSV) isolate in this study was isolated from tissues of a subclinical Stimson’s python (*Antaresia stimsoni*) that resided in a colony of mixed snakes, including multiple *Antaresia, Python*, *Morelia*, and *Boa* species. Growth of the virus was confirmed using the modified rtPCR PVTF/GGTR protocol described above in first passage lysate. AnPSV inoculum used for characterization was unfiltered lysate of a passage 4 preparation. DPHt inoculated with AnPSV lysate were frozen when cells in the flasks showed peak CPE at approximately four to five days post-inoculation. Sanger sequencing placed the closest match to this isolate as a blood python serpentovirus with Genbank accession number MN161565. 

### 2.6. Establishing a Median Tissue Culture Infectious Dose (TCID_50_) Assay

A median tissue culture infectious dose (TCID_50_) assay was developed to establish the infectious titer of serpentoviral samples. Each well of a 96-well plate (Genclone) was seeded with 2.5 × 10^4^ DPHt cells suspended in 100 μL of completed growth medium, and plates were placed in a humidified, 32 °C incubator supplemented with a 5% CO_2_ atmosphere. After approximately 24 h, DPHt cells formed a confluent monolayer. Serial 1:1 dilutions of viral lysate were made down to 1:8.8 × 10^12^. The growth medium was aspirated from all DPHt monolayers, and each well was inoculated with 25 μL of the serially diluted virus. All samples were run in triplicate. For each replicate, a sham-inoculated well (growth medium alone) was used as a control. Viral adsorption was permitted for 60 min at room temperature (20 °C), after which 100 μL of additional growth media was added to each well. Plates were returned to a 32 °C, humidified 5% CO_2_ atmosphere incubator for five days. The TCID_50_ was calculated using the Spearman-Karber method [25] following determination of the greatest sample dilution resulting in at least a 50% decrease in cell number compared to the sham-inoculated well. The TCID_50_ measurements were log-transformed, and the replicates for each sample were averaged for comparison. All graphs of TCID_50_ measurements were performed in Rstudio v2021.09.2 (https://rstudio.com with package ggplot2 (Version 3.3.5)).

### 2.7. Freeze-Thaw Characterization

Viral stock flasks (frozen and stored at −80 °C) were thawed and 200 μL was aliquoted per PCR tube. This initial thaw constituted the first freeze-thaw event. For additional freeze-thaw treatments, a subset of samples was returned to −80 °C until completely frozen, and the process was repeated up to three additional times. The infectious titer of controls and experimental samples was determined by TCID_50_ as previously described. 

### 2.8. Environmental Stability

Serpentoviral environmental stability was assessed for five temperature points: −20 °C, 4 °C, 20 °C (Room Temperature), 32 °C, and 35 °C. Viruses were aliquoted as for the freeze-thaw experiment, samples were stored at each respective temperature, and time points were collected every two days for 22 days. All samples were stored at −80 °C until the completion of the experiment, at which time the infectious titers were assayed in triplicate by TCID_50_ as previously for each temperature, timepoint, and virus combination. Stock virus was used as an untreated control. 

### 2.9. Single-Cycle Growth Kinetics

Single-cycle growth kinetic (SCGK) experiments were performed using three cell lines: DPHt, African green monkey kidney (Vero), and corn snake mass (CoSnMs) cell lines. Cells were grown in T12.5 flasks containing 2 mL of growth media until confluent. The growth medium was removed, and monolayers were inoculated with the virus at a multiplicity of infection (MOI) of 5 (as determined on DPHt cells) diluted in 400 μL of growth medium. To reach the target MOI, a TCID_50_ was calculated for viral stock, and the number of DPHt cells in a representative flask or well was counted by hemocytometer. Viral stock was diluted with cell culture media to reach the appropriate MOI based on the observed number of cells. Virus adsorption was permitted for 1 h at room temperature, 2 mL of growth medium was added to each flask, and flasks were returned to a 32 °C incubator with a humidified 5% CO_2_ atmosphere. For the first 96 h (Days 0–4), flasks were collected every four hours. For hours 96–240 (Days 4–10), flasks were collected every twelve hours. All flasks were stored frozen at −80 °C until the completion of the experiments. Flasks were subjected to three freeze–thaw cycles to lyse cells prior to being assayed in triplicate by TCID_50_ as previously described. 

To compare viral replication at an elevated temperature (35 °C), the procedure above was repeated on DPHt cells acclimated to 35 °C growth. The procedure was the same as above, though time points were collected once every 24 h. Flasks were subjected to three freeze–thaw cycles to lyse cells prior to being assayed in triplicate by TCID_50_ as previously described. 

### 2.10. Multiple-Cycle Growth Kinetics

Multiple-cycle growth kinetics were determined on the DPHt cell line. Conditions were similar to those described for 32 °C SCGK experiments above, only flasks were inoculated with the virus at a MOI = 0.1. Flasks were subjected to three freeze-thaw cycles to lyse cells prior to being assayed in triplicate by TCID_50_ as previously described.

### 2.11. Epifluorescent Immunostaining to Assess Infection

Epifluorescent immunostaining and fluorescent microscopy were used to assess cellular infection in vitro. Indirect fluorescent antibody screening was performed using a previously characterized polyclonal antibody targeting the serpentovirus nucleocapsid protein [8]. 

To quantify serpentoviral infection of DPHt cells at 32 and 35 °C, a glass coverslip was placed into each well of a 12-well plate (Genclone), and each well was seeded with 2.0 × 10^5^ cells. Once a confluent monolayer was observed growing over the coverslip, wells were inoculated with 200 µL of either serpentovirus at an MOI = 5 or straight growth medium (mock). Virus adsorption was permitted for one hour at room temperature before each well was topped with 1000 μL of growth medium and returned to a 32 or 35 °C, humidified 5% CO_2_ atmosphere.

A panel of tissue culture cell lines from various species was used to assess for in vitro host range potential. The panel tested included 32 cell lines in total, including nine boid snake cell lines, four colubrid snake cell lines, three viper snake cell lines, three lizard cell lines, four chelonian cell lines, three crocodilian cell lines, three avian cell lines, and three mammalian cell lines. The species, origin tissue, and commercial name (if applicable) for each cell line utilized in this study are shown in Table 1. Commercial cell lines and budgie cell line [26] were shared for use in this study. Prior to inoculation, all reptile cell lines were maintained in a 32 °C, humidified 5% CO_2_ atmosphere with completed media. Mammalian and avian cell lines were acclimated to both grow at 32 °C as well as to the previously described growth, except for the Japanese Quail Fibrosarcoma (QT-35) cell line, which was grown in Dulbecco’s Modified Eagle Medium (DMEM; GenClone) with 5% Heat Inactivated fetal bovine serum (FBS; GenClone). Cells were seeded into a 12-well plate containing glass coverslips. Once a confluent monolayer was observed growing over the coverslip, wells were inoculated with 200 µL of either serpentovirus at an MOI of 5 (as determined on DPHt cells) or straight growth medium (mock). Virus adsorption was permitted for one hour at room temperature before each well was topped with 1000 μL of growth medium and returned to a 32 °C, humidified 5% CO_2_ atmosphere.

For all immunofluorescent studies, 48 h post-inoculation, media were removed from each well, and the wells were washed twice with phosphate-buffered saline (PBS). Cells were fixed with 1000 μL of 2% paraformaldehyde (PFA) for 1 h and washed twice with PBS. Fixed coverslips were stored in PBS at 4 °C until staining. 

The immunostaining protocol using rabbit anti-NdvNcAb was performed as described in Hoon-Hanks et al. (2018), with the following modifications: 1% normal goat serum (Thermo Fisher) was used in replacement of 1% bovine serum. A secondary Alexa Fluor 594 goat anti-rabbit IgG antibody was used in replacement of Alexa Fluor 488 goat anti-rabbit IgG antibody; and DAPI (1:10,000; Thermo Scientific) was used in replacement of Hoeschst 22242 for DNA nuclear staining. Following completion of immunostaining, the coverslips were mounted on glass microscope slides with Fluoromount-G (Thermo Fisher) and stored in the dark at 4 °C until examination. Cells were imaged ×600 magnification on a Nikon Eclipse Ci microscope taken with NIS Elements D software (Version 5.30.05). Images were processed in GIMP 2.10.30 (2017), and both infected and mock wells were processed analogously to reflect observed fluorescent intensities.

To quantify infection of DPHt cells at 32 and 35 °C, three representative images were taken for each temperature–virus combination ×400 magnification. Infected cells and cell nuclei were counted in the captured images using DotDotGoose (Version 1.6.0) [27]. The association between infected cell counts and temperature was determined with a Chi-squared analysis with an alpha of 0.05. 

### 2.12. Sanitizer Efficacy

Five sanitizers commonly used in veterinary medicine and captive herpetoculture were tested for their ability to neutralize serpentoviruses: 7% peroxide, 10% consumer sodium hypochlorite, 70% ethanol, 0.4% quaternary ammonium cleaner (F10 SC, F10 Products, Roodepoort, South Africa), and 2% chlorohexidine gluconate (4% Chloradine Scrub, Aspen Veterinary Resources, Liberty, MO, USA). The cytotoxicity of each sanitizer on DPHt cells was determined by serially diluting the chemicals on monolayers in 96 well plates as used for the TCID_50_ assay. The dilution of sanitizer that did not cause an observable cytopathic effect was used as the starting dilution post-sanitizer treatment for all samples prior to TCID_50_ quantification.

Serpentoviruses were mixed with concentrated sanitizer to reach the manufacturer’s recommended neutralizing dilution. Other controls run simultaneously included a virus-only treatment (virus diluted with growth medium to a level equivalent to the neutralizing dilution) and a sanitizer-only treatment (sanitizer diluted with growth medium to a level equivalent to the neutralizing dilution). Each control treatment was run singularly, while experimental treatments were run in duplicate. Samples were incubated for one or three minutes and then further diluted with growth medium to reach the previously determined safe inoculation dilution. Residual infectivity was quantified by TCID_50_ as previously described.

### 2.13. Antiviral Test Compounds

Ribavirin was obtained from ICN Pharmaceuticals, Inc. (now Bausch Health, Laval, QC, Canada). Favipiravir was purchased from AdooQ Bioscience (Irvine, CA, USA). NITD-008 was supplied by BEI Resources (Manassas, VA, USA). Enviroximine was purchased from Cayman Chemical (Ann Arbor, MI, USA), and pirodavir was purchased from Med Chem Express (Monmouth Junction, NJ, USA). Infergen (Interferon-alfacon-1), a human consensus interferon, was provided by InterMune, Inc (Brisbane, CA, USA) as an aqueous solution. Solid compounds were dissolved in DMSO at a 20× concentration to yield a final DMSO concentration of <10%. Further dilutions were performed in culture media.

### 2.14. Antiviral Efficacy

A CPE reduction assay was used to evaluate the effect of ribavirin, remdesivir, favipiravir, NITD-008, enviroximine, pirodavir, and infergen against AnPSV, BPNV, and GTSV. Diamond python heart cells (DPHt) were grown as described above. Cells were plated in 96-well microtiter plates and grown overnight prior to the addition of the virus and test compounds. A CPE assay was conducted to determine the activity of various antiviral against BPNV, GTSV, and AnPSN. Serial half-log dilutions of the test compounds were made and added to 5 wells of the 96-well plate. Culture media was added to two wells to determine the 50% cytotoxic concentration (CC50), and the virus was added to the other three wells to determine the 50% effective concentrations (EC50). Virus and cell controls were included with each plate. The selective index was determined by dividing the CC50 by the EC50/90 to determine the SI50/90. Two to three replicates of the assays were used to evaluate each compound. A confirmatory virus yield reduction (VYR) assay was conducted where supernatant from the CPE assay was collected and titered on DPHt cells to determine the 90% effective concentration (EC90), or the concentration of compound required to reduce the virus titer by one Log_10_.

## 3. Results

### 3.1. Freeze-Thaw Stability

All three isolated serpentoviruses, Ball Python Nidovirus (BPNV), Green Tree Python Serpentovirus (GTSV), and Antaresian Python Serpentovirus (AnPSV) showed similar, relatively low declines in infectivity following repeated cycles of freezing and thawing (Figure 1A). While, on average, GTSV had the largest decrease in infectious titer following each freeze–thaw cycle (average loss of infectivity of −0.43 ± 0.13 SE Log_10_ TCID_50_/mL), all viruses were similar and showed a decrease of approximately 0.5–1 Log_10_ infectivity following the third freeze/thaw iteration.

### 3.2. Environmental Stability

All tested serpentoviruses exhibited a pattern of decreasing infectivity following storage at increasing temperatures (Figure 1B). At 35 °C, serpentoviruses showed a complete loss of infectivity by 8–10 days, representing a net loss of infectivity of ~7–10 Log_10_ TCID_50_/mL. Although viral infectivity persisted for up to 10 days at 35 °C, all isolates showed a 65 ± 13% SE (4.29 ± 0.47 SE Log_10_ TCID_50_/mL, BPNV) to 100 ± 0% SE (9.18 ± 0.00 SE Log_10_ TCID_50_/mL, GTSV) reduction in infectivity within two days. Similarly, at 32 °C serpentoviruses retained infectivity for 8–10 days, though infectivity reductions of only 20 ± 0% SE (1.37 ± 0.00 SE Log_10_ TCID_50_/mL, BPNV) to 68 ± 2% (8.28 ± 0.10 SE Log_10_TCID_50_/mL, GTSV) was observed within two days. At room temperature (20 °C), all detectable infectivity was lost for the three in 10–12 days, but only a 24 ± 5% SE (2.42 ± 0.47 SE Log_10_TCID_50_/mL, AnPSV) to 60 ± 2% SE (5.48 ± 0.10 SE Log_10_TCID_50_/mL, GTSV) reduction in infectivity after two days.

Viruses showed the greatest stability at colder temperatures, with retention of some degree of infectivity at both −20 °C and 4 °C after 22 days (Figure 1B). At 4 °C, a maximum of 38 ± 7% SE (3.46 ± 0.37 SE Log_10_ TCID_50_/mL, GTSV) to 41 ±14% SE (2.72 ± 0.37 SE Log_10_TCID_50_/mL, BPNV) decrease in infectivity was documented during the experiment. At −20 °C, a maximum of 25 ± 3% SE (2.29 ± 0.17 SE Log_10_TCID_50_/mL, GTSV) to 30 ± 6%SE (3.01 ± 0.57 SE Log_10_TCID_50_/mL, AnPSV) infectivity loss was observed.

### 3.3. Growth Kinetics

To make comparisons of viral growth rates between the three serpentovirus isolates, single-cycle kinetics of viral growth (MOI = 5) were determined for the three serpentoviruses, two reptile cell lines (DPHt and CoSnMs), and a primate cell line (Vero) (Figure 2A,C,D). DPHt cells were established heart tissue of a *Morelia s. spilota*, a competent host species for Pregotovirus-genus serpentoviruses [11], and have been successfully used to isolate serpentoviruses in this and previous studies [8]. CoSnMs cells were established from tissues of *Pantherophis guttatus*, a snake not documented as a host for serpentoviruses in the genus *Pregotovirus*, but that can support infection by more divergent serpentoviruses [14]. Lastly, Vero cells are an immune-deficient primate cell line frequently used for virus isolation. To simplify data presentation, only data points at 8 h intervals during the first 96 h are shown in Figure 2C,D for single-cycle growth in CoSnMs and Vero cells. The kinetics of multiple cycles of growth (MOI = 0.1) of all three viruses on DPHt cells was also performed (Figure 2B).

On DPHt cells under single-cycle growth conditions, all serpentoviral titers began to increase within ~8 h post-inoculation (hpi). The greatest increase in titer occurred between 8–24 hpi. However, virus titers did not peak until 40–48 h (Figure 2A). The greatest increase in titer for each virus was as follows (Log_10_ TCID_50_/mL): AnPSV, 7.08 ± 0.99 SE; BPNV, 5.58 ± 0.10 SE, and GTSV, 4.32 ± 0.00 SE.

In contrast, no net increase in infectivity was noted in CoSnMs cells inoculated with each of the three tested viruses. For all, a decrease in infectious titer over control samples was noted for the first 24–48 h (Figure 2C), likely representing a period of potential viral adsorption and entry rending viral particles non-infectious. For GTSV, titers declined steadily for the first 24 hpi but then began showing viral growth until peaking at 96 h, though still not reaching the baseline inoculum titer (Figure 2C). Both BPNV and AnPSV showed absent to minimal evidence of productive replication on CoSnMs cells. For AnPSV at 48 hpi (also potentially minimally at 64 and 72 hpi), a minor titer elevation compared to the previous time point(s) was captured. 

In Vero cells inoculated with serpentoviruses at a MOI = 5, a net increase in infectious titer over time was documented, though the kinetics of single-cycle growth were delayed when compared to growth on DPHt cells (Figure 2D). The highest titers were reached by BPNV (4.65 ± 0.10 SE Log_10_TCID_50_/mL, 168 hpi). Both GTSV (2.19 ± 0.50 SE Log_10_TCID_50_/mL, 96 hpi) and AnPSV (1.33 ± 0.33 SE Log_10_TCID_50_/mL, 108 hpi) showed more modest maximal growth titers. 

Under multiple-cycle growth conditions (MOI = 0.1) on DPHt cells, all three viruses achieved a higher peak viral titer (Figure 2B) compared to single-cycle growth (Figure 2A). All three viruses showed similar growth rates for the first 48 h. While the multiple-cycle growth of both BPNV and GTSV peak at 48 h, AnPSV-infected cells continue to produce infectious particles until 60 hpi (Figure 2B). In addition to a slightly lengthened period of multiple-cycle growth on DPHt cells, AnPSV also achieved a much higher overall increase in infectious titer (13.92 ± 0.27 SE Log_10_TCID_50_/mL) compared to both BPNV (8.81 ± 0.57 SE) and GTSV (9.08 ± 0.17 SE).

### 3.4. Temperature-Dependent Growth Kinetics

To examine the potential thermal limitations of serpentoviral growth, single-cycle growth kinetics were assessed on DPHt at 35 °C. In contrast to the growth curves noted at 32 °C, a complete loss of viral infectivity was noted 24–48 hpi at 35 °C (Figure 3). At corresponding time points of single-cycle growth kinetics at 32 °C on DPHt cells, the viruses were attaining peak infectious titers. 

To further investigate the potential for temperature-restricted serpentoviral infection, epifluorescent immunostaining using a polyclonal anti-nucleocapsid antibody was performed to compare infection rates 48 hpi on DPHt cells at 32 °C and 35 °C (Table 2, Figure 4A–H). While between 11.6% (122/1053 cells, BPNV) to 27.9% (266/953 cells, AnPSV) of cells were infected at 32 °C, increasing the incubation to 35 °C resulted in drastic reductions in infectivity, with only 0.4% (3/694 cells, GTSV) to 0.5% (4/805 cells, AnPSV) of cells being immunopositive (Table 2). This difference was statistically significant via Chi-squared test for all isolates (*p* < 2.2 × 10^−16^). The disparity between infection rates at 32 °C versus 35 °C can be visualized in the representative field images in Figure 4A–H. 

### 3.5. In Vitro Host Range Assessment by Epifluorescent Immunostaining

A panel of 32 tissue culture cell lines from a diversity of taxa, including reptiles, birds, and mammals, was used to assess an in vitro host range for the three isolated serpentoviruses. The panel consisted of nine boid (boa and python) snake lines representing six species/subspecies, four colubrid snake lines from four species, three viperid snake lines from three species, three lizard cell lines from three species, four chelonian (turtles and tortoises) lines from four species, three crocodilian (alligators, caiman, and crocodiles) lines from three species, three avian lines from three species, and three mammalian lines from three species (Table 1). 

BPNV, GTSV, and AnPSV each showed a wide in vitro host range, with successful viral replication as detected by cytoplasmic nucleocapsid protein production recorded in all 32 tested cell lines, representing multiple taxonomic groups and organ systems (Table 1, Figure 4, Figure 5 and Figure 6). The pattern of immunostaining varied by the virus and cell line. In a minority of cells, immunofluorescence was restricted to apparent cellular organelles (Figure 4V–X,Z–BB, Figure 5V–X and Figure 6J–L,O,P). However, some cell line-virus combinations resulted in diffuse cytoplasmic immunostaining (Figure 4J–L and Figure 5H,N,T). In cell line-virus combinations that exhibited a high degree of CPE, such as DPHt, intense immunofluorescence surrounded the nuclei in balled-up cells (Figure 4F–H). Though immunostaining was noted in all cell lines inoculated with serpentoviruses, there was variation in the fluorescence intensity of immunostaining. In Burmese python heart cells, BPNV- and GTSV-infected cells showed strong immunofluorescence (Figure 4J,K), while AnPSV-infected cells exhibited notably weaker immunostaining (Figure 4L). This pattern was not consistent, however, and no common pattern as it relates to immunostaining intensity by different viruses was observed. For example, in Vero cells, BPNV-infected cells exhibited strong immunostaining (Figure 6J), while GTSV- and AnPSV-infected cells exhibited weaker immunostaining (Figure 6K,L).

### 3.6. Sanitizer Efficacy

To address the efficacy of commonly used sanitizers to inactivate serpentoviruses, a panel of five disinfectants was tested after one and three minutes of contact time. All three isolates were significantly inactivated by all five tested disinfectants (Figure 7). One minute contact time was sufficient to completely (100%) neutralize the infectious potential of the tested serpentoviruses for 10% bleach, 70% ethanol, and 2% chlorhexidine. The quaternary ammonium compound (QAC) was able to neutralize 100% of AnPSV and GTSV in one minute, while BPNV was reduced by 95 ± 11% SE. All viruses were completely neutralized after three-minute contact time with QAC. The sanitizer with the lowest efficacy was 7% peroxide, with complete neutralization in three minutes, but reductions in infectivity of only 13 ± 29% for BPNV to 54 ± 39% for AnPSV after one-minute contact time. 

### 3.7. Antiviral Efficacy

A CPE reduction assay was used to evaluate the effect of ribavirin, remdesivir, favipiravir, NITD-008, enviroximine, pirodavir, and infergen against AnPNV, BPNV, and GTSV (Table 3). Remdesivir (RDV), ribavirin (RIBA), and NITD-008 (NITD) were found to have potent antiviral activity against all three serpentoviruses with SI50s ranging from 39 to 1100 depending on compound and virus. Activity was confirmed using a virus yield assay (VYR). The other compounds showed low or no activity.

## 4. Discussion

The results of this study represent the first in vitro characterization of serpentoviruses (reptile-associated nidoviruses). Serpentovirus infections are best documented in captive snake populations [11,15,16]. These viruses can be detected at a high prevalence in captive python populations and can cause outbreaks of considerable economic and clinical significance [10,11,15,28]. Generated data regarding viral environmental stability, disinfectant susceptibility, and in vitro host range will prove helpful in mitigating the spread of serpentoviruses in captivity. Serpentoviruses are also documented in wild reptile populations [9,10,14], and results from this study could inform field technicians working with vulnerable wild reptile populations as it relates to biosecurity.

Most current recommendations to mitigate the spread of serpentoviruses in captivity are focused on limiting viral transmission via fomites. Results from the three serpentoviruses characterized in this study indicate environmental stability similar to other RNA-enveloped viruses, remaining infectious for up to 10 to 12 days in the environment at room temperature (20 °C). The period of environmental infectivity decreased at higher temperatures and increased at cooler temperatures. These findings are consistent with other nidovirales, such as Turkey coronavirus (TCoV), human coronavirus HCoV-229E, and SARS-CoV-2, which can last in suspension up to 10, 9, and 7 days, respectively, at room temperature, but much longer at cooler temperatures [17,18,19].

Prior to this study, any recommendations for serpentoviral disinfection were based on related, surrogate viruses. However, those working with snakes in captivity can now reliably reach for a number of proven disinfectants to limit virus transmission. Serpentoviruses showed a wide susceptibility to all the sanitizers used in the study as 10% bleach, 70% ethanol, 2% chlorhexidine, and QAC fully or nearly fully neutralizing the virus after only 1 min. Although 7% peroxide solution was significantly less effective than the other sanitizers with a one-minute contact time, exposure for three minutes fully inactivated the tested viruses. The sensitivity to a variety of widely available and commonly used disinfectants following a very brief contact time means that disinfection in practically all captive situations should be easily achievable with appropriate biosecurity and cleaning protocols. The disinfectant sensitivity for serpentoviruses, much like the environmental stability, is also similar to other nidoviruses such as SARS-CoV-2, which have proven susceptible to a wide variety of sanitizers, including 10% bleach, 75% ethanol, QAC, 0.05% chlorhexidine, and hydrogen peroxide solutions [19,20,21].

Characterization of in vitro growth kinetics across three cell lines yielded a number of findings as it related to viral proliferation. In a python-origin cell line, all three viruses exhibited rapid and substantial expansion. Peak viral titers in single-cycle growth conditions were reached in 24–48 h and in 48–60 h in multiple-cycle conditions, indicating the potential for rapid viral proliferation in infected cells. The viruses were also able to produce large amounts of infectious progeny in the python cell line, with 4–7 log_10_ increases for each virus under single-cycle conditions and 9–14 log_10_ increases under multiple-cycle conditions. This replication capacity may partly explain why these viruses can be so problematic in captivity, as they have the potential to produce heavy viral loads to both disseminate within the host as well as seed the environment. The variation in productive viral yield may also explain why predicting the clinical outcome for any specific infection is challenging [11]. However, only three cell lines were used to assess viral growth, and only one of those cell lines was from a “susceptible” snake species, DPHt. It is possible that growth kinetics on a variety of cell lines of varying tissues may provide more insight into the possible relations between growth kinetics and disease in pythons.

In contrast to the findings in the python cell line, a colubrid snake cell line (CoSnMs) did not generate a net positive change in infectious titer. This is intriguing from multiple fronts. While a serpentovirus has been documented in a free-ranging corn snake, that snake did not exhibit clinical signs [14]. Moreover, no spillover of python-associated serpentoviruses has been detected in a number of retrospective molecular studies [11,15]. This suggests that there may be restrictions at the cellular level as it relates to successful python serpentoviral infection in colubrid snakes. This is in part supported by the immunofluorescence finding that CoSnMs cells are successfully infected by the tested virus, but apparently, either infected cells produce limited amounts of infective virus, or the rate of cellular infection is too low to produce high levels of infectious particles. 

Two particularly powerful findings from this study relate to the identification of potential therapeutic options, both pharmacological and non-pharmacological, for snakes with serpentovirus-associated disease. One of the primary challenges of dealing with serpentoviral infections in captive snakes from an animal health perspective is a lack of appropriate treatment regimens. Currently, snakes with serpentoviral disease are predominantly treated with supportive care. However, of the seven antiviral compounds examined, three (RDV, RIBA, and NITD) showed promise in restricting serpentovirus growth. Ribavirin has been used clinically for decades for the treatment of various human viral diseases, including respiratory syncytial virus and Hepatitis C virus (HCV), although long-term treatment with this compound is associated with anemia which limits its utility [29]. Remdesivir was used in several countries during the COVID-19 pandemic [30] and has broad-spectrum activity against a wide variety of viruses in vitro and in vivo [31,32,33]. NITD008 showed promise against the Dengue virus in animal models, but development ceased after toxicity was observed in rats and dogs [34]. These compounds could therefore be potentially useful if they prove safe and protective in an in vivo system. 

The finding of in vitro temperature-restricted productive infection of serpentoviruses may provide another potential treatment option, one that avoids the stress typically associated with the administration of medications in exotic species. While serpentoviral growth could not be sustained at 35 °C in culture, this temperature resides within the thermal tolerance for snakes [35] and care recommendations of some python species [36,37]. A number of species of Australian pythons, including the water python (*Liasis fuscus*), amethystine python (*Simalia amethistina*), black-headed python (*Aspidites melanocephalus*), and carpet python (*Morelia spilota*) have been documented voluntarily heating up a body temperature maximum of approximately 35 °C in the wild [38,39,40]. Physiology studies of captive pythons have heated pythons to temperature maximums of 35 to 40 °C for several days without reported incidents [41,42]. A treatment regimen involving heating infected snakes to temperatures intolerant to serpentovirus growth (35 °C) could prove a potential treatment option. Temperature modifications to treat infectious diseases are done for both domestic and non-domestic species. In dogs, increasing the ambient temperature is a strategy sometimes employed to treat canine herpesvirus infections in neonatal puppies [43]. In frogs with the fungus *Batrachochytrium dendrobatidis*, heat treatment has shown therapeutic promise in multiple frog species [44,45,46]. Although further studies applying these in vitro findings to live-animal models are needed before making general clinical recommendations, the findings of this study show promise for potential therapeutics for an otherwise untreatable disease. 

The ability of the three serpentoviruses used in this study to infect a diverse panel of cell lines, including a variety of reptiles, birds, and even mammals, was unexpected. Such host plasticity has not been observed in vivo, as documented serpentovirus infections are largely limited to captive, and to a lesser degree free-ranging, boids and a handful of other reptile taxa [9,10,11,13,14]. Even within snakes, there seem to be limited viral susceptibility and transmission between families. The more restricted host range observed in vivo may be due to a number of factors. As shown in our growth kinetic experiments, serpentovirus replication is limited by elevated temperature (35 °C). The physiologic temperature of most mammals and birds is greater than 35 °C, which may limit successful or productive infection of these hosts in vivo. In possible support of this, productive serpentoviral infection was noted in Vero cells adapted to 32 °C but was absent at 35 °C. Other reptile viruses have also shown wide in vitro host ranges not reflected in vivo, including reptarenaviruses, which can infect a variety of mammalian and reptilian cell lines [47], and lizard rhabdoviruses which can infect mammalian, reptilian, and amphibian cell lines [48]. Another factor that may also limit the serpentoviral in vivo host range is the role immune defenses play in preventing viral infection [49,50]. Finally, other factors, such as the production of subgenomic RNA and other ‘accessory proteins’ by nidoviruses, may not be necessary for replication in vitro but can play a critical role in pathogenesis for in vivo infections [51,52]. Ultimately, however, additional studies specifically screening for evidence of serpentoviral infections in mammals and birds are necessary to assess their infective potential. 

Moreover, even within this study, positive serpentoviral immunostaining did not necessarily equate to detectable productive infection as assessed by TCID_50_. While individual CoSnMs cells exhibited positive immunoreactivity for all three isolated serpentoviruses, single-cycle growth kinetic experiments showed that only GTSV infection consistently yielded infectious titers. Conversely, although small numbers of DPHt cells grown at 35 °C demonstrated positive immunoreactivity for all three serpentoviruses (Figure 4B–D), no isolate was able to productively infect the same cells at this temperature as assessed by TCID_50_.

While the wide in vitro host range may have limited clinical significance, the broad in vitro capacity for infection may indicate that one or more cellular receptors bound by the serpentovirus spike protein is either a highly conserved protein target or a ubiquitous non-proteinaceous moiety. The related nidovirus SARS-CoV-2 targets the highly conserved and broadly expressed protein receptor angiotensin I converting enzyme (ACE2), resulting in a wide cellular and tissue tropism and a broad in vivo host range [53,54]. Furthermore, evidence of past spillover events between reptile taxa can be found in serpentovirus phylogeny, which includes chameleon viruses [13], turtle viruses [9], and skink viruses [10] mixed within the otherwise python virus dominated serpentovirus genus *Pregotovirus* [14]. Although these spillover events may occur infrequently, the low-level replication of GTSV in CoSnMs cells suggests sustained viral growth in divergent taxa may be possible. 

Though the data generated in this study, providing in vitro characterization of the stability, growth, and host range of isolated serpentoviruses, was entirely in vitro in nature, the results give promising directions for future study as it relates to disease development, clinically susceptible host range, and potential therapeutic treatments. While further in vivo experiments would be needed to determine the safety and efficacy of any therapeutic treatments, the reported results provide, for the first time, a framework that can be used to mitigate the effects of these viruses in captive snakes.

## Figures and Tables

**Figure 1 microorganisms-11-01371-f001:**
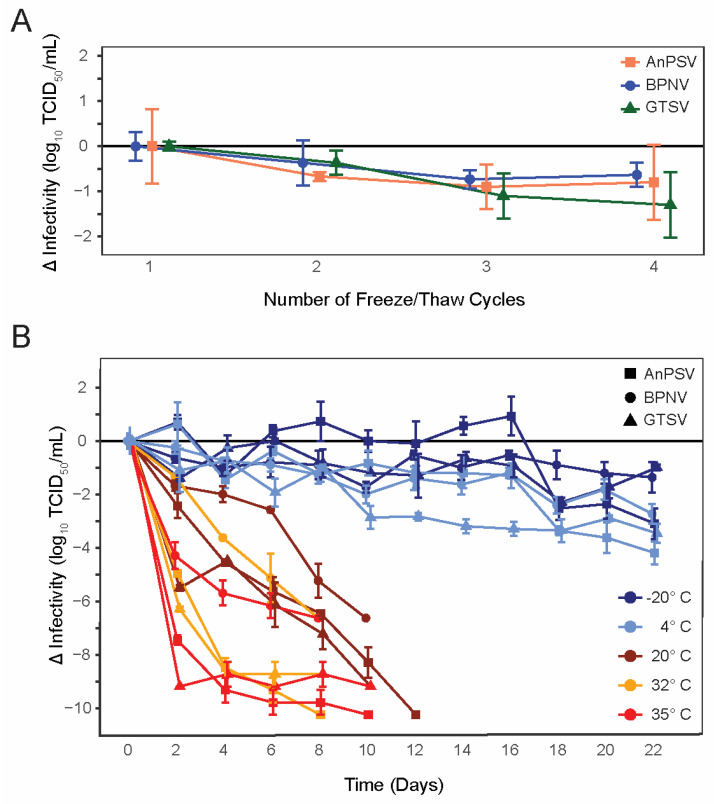
Serpentoviral freeze/thaw and environmental stability. (**A**) Changes in infectious titer of three isolated serpentoviruses following repeated freeze–thaw cycles as determined by median tissue culture infectious dose (TCID_50_) assay. Standard error bars of triplicate measurements are shown. (**B**) Changes in infectious titer of three isolated serpentoviruses isolated serpentoviruses following incubation at one of five different temperatures: −20, 4, 20, 32, and 35 °C. Additional data points for samples showing a complete loss of infectious titer before the conclusion of the experiment have been omitted.

**Figure 2 microorganisms-11-01371-f002:**
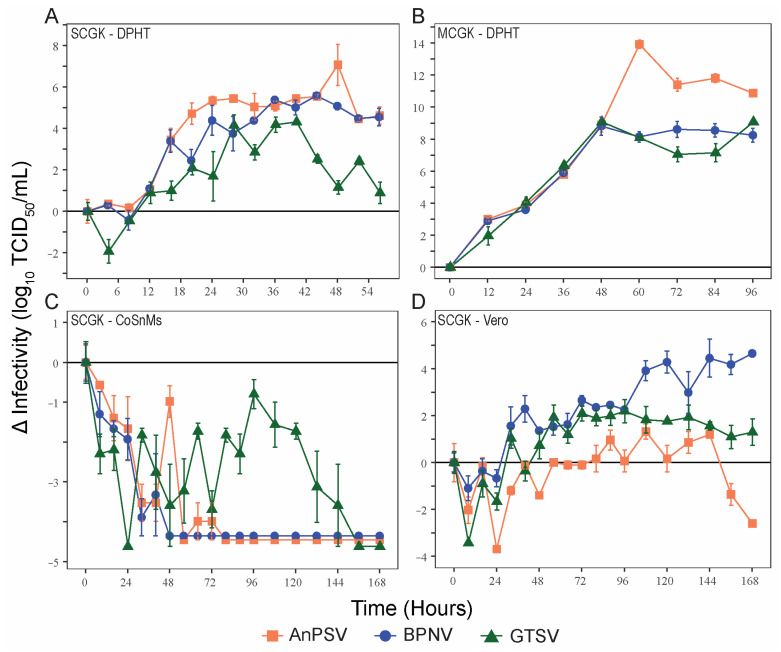
Serpentoviral in vitro growth kinetics. Growth kinetics of three isolated serpentovirus on diamond python heart (DPHt; (**A**,**B**)), corn snake mass (CoSnMs; (**C**)), and green monkey kidney (Vero; (**D**)) cell lines. Monolayers of the respective cell lines were inoculated with serpentovirus at an MOI = 5 to assess single-cycle growth (**A**,**C**,**D**) and MOI = 0.1 for multi-cycle growth (**B**). Standard error bars representing averaged triplicate measurements are shown for each data point. Additional data points extending beyond the growth kinetic curves have been omitted for clarity.

**Figure 3 microorganisms-11-01371-f003:**
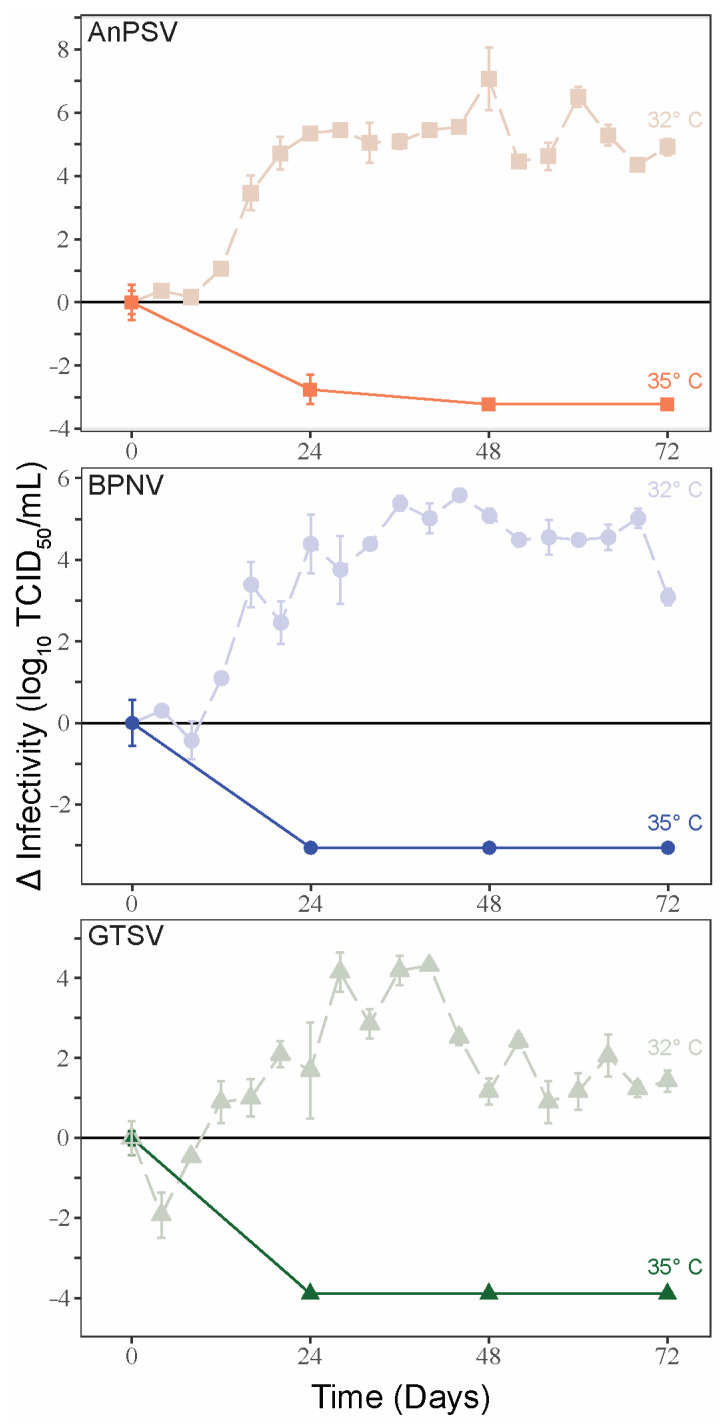
Temperature-restricted growth of serpentoviruses. Single-cycle growth kinetic experiments were performed for the three serpentoviruses on Diamond Python Heart (DPHt) cells at 35 °C, with standard error bars from averaged TCID_50_ measurements run in triplicate. Data from single-cycle growth kinetic experiments at 32 °C is included on each graph for comparison.

**Figure 4 microorganisms-11-01371-f004:**
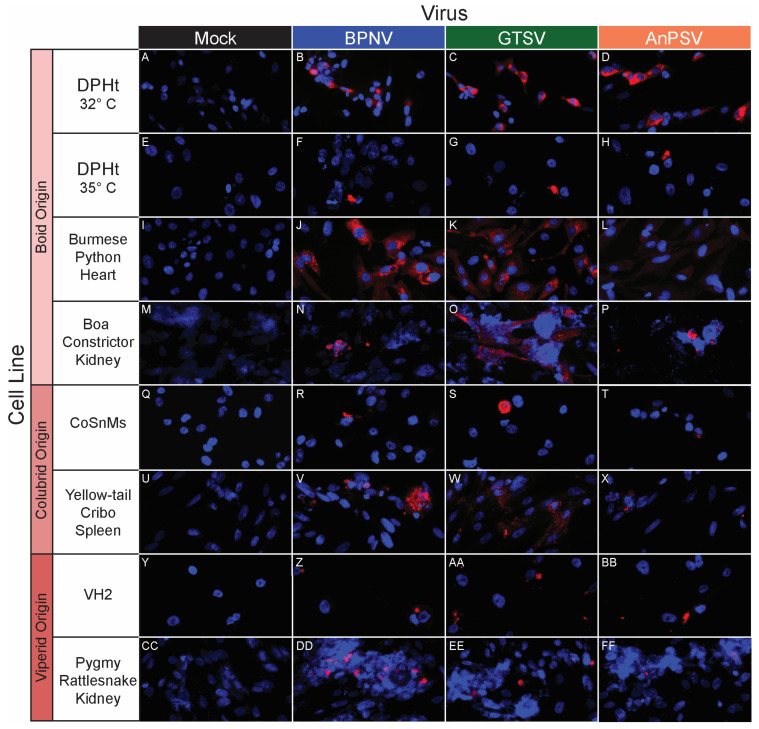
In vitro serpentoviral host range of snake cell lines as assessed by epifluorescent immunostaining. A panel of snake cell lines including Boid (**A**–**P**), Colubrid (**Q**–**X**) and Viperid (**Y**–**FF**) cell lines was inoculated and incubated with three serpentovirus isolates for 48 h, followed by fixation and immunostaining. Viral nucleocapsid protein (red) and DAPI nuclear staining (blue) are shown at 600× magnification. Incubations were performed at 32 °C for all except panels (**E**–**H**), which were incubated at 35 °C. Commercial cell lines are identified by their abbreviations.

**Figure 5 microorganisms-11-01371-f005:**
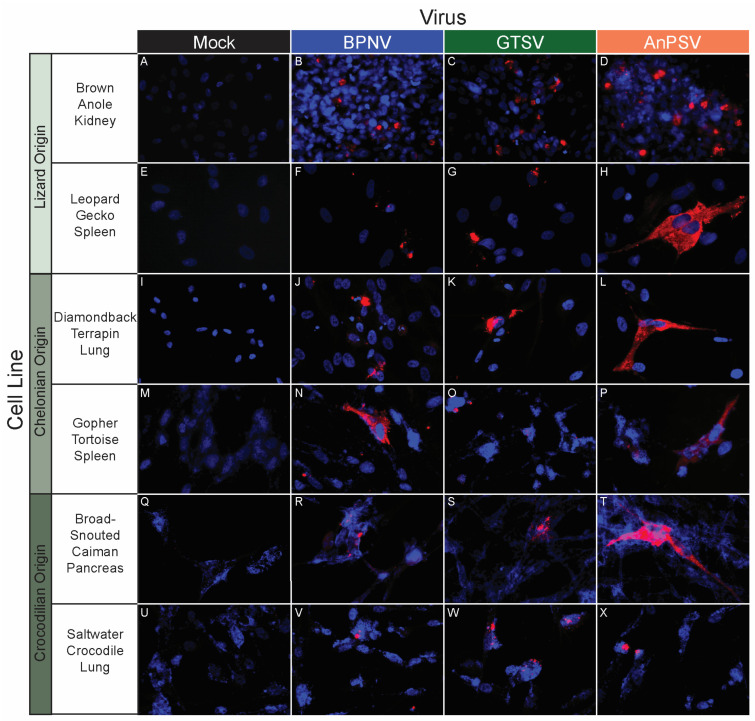
In vitro serpentoviral host range of other reptile cell lines as assessed by epifluorescent immunostaining. A panel of reptile cell lines, including representatives from lizards (**A**–**H**), chelonians (turtles and tortoises, **I**–**P**), and crocodilians (alligators, caiman, and crocodiles, **Q**–**X**) was inoculated and incubated with three serpentovirus isolates for 48 h, followed by fixation and immunostaining. Viral nucleocapsid protein (red) and DAPI nuclear staining (blue) are shown at 600× magnification. Incubations were performed at 32 °C for all.

**Figure 6 microorganisms-11-01371-f006:**
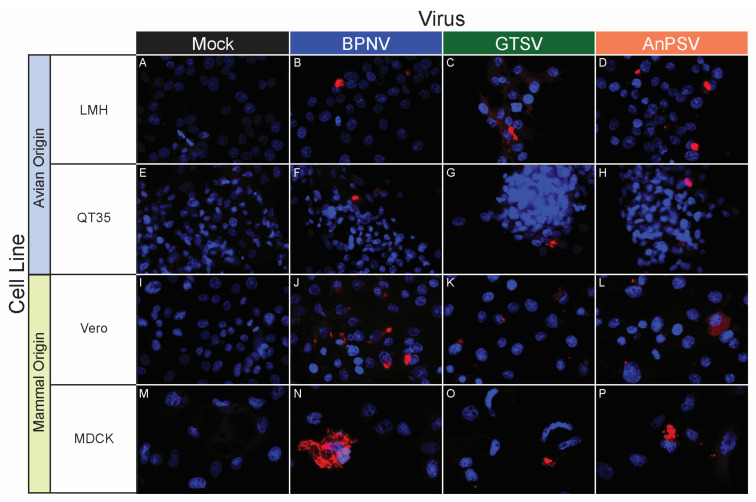
In vitro serpentoviral host range of avian and mammal cell lines as assessed by epifluorescent immunostaining. A panel of avian (**A**–**H**) and mammalian (**I**–**P**) cell lines was inoculated and incubated with three serpentovirus isolates for 48 h, followed by fixation and immunostaining. Viral nucleocapsid protein (red) and DAPI nuclear staining (blue) are shown at 600× magnification. Incubations were performed at 32 °C. Commercial cell lines are identified by their abbreviations.

**Figure 7 microorganisms-11-01371-f007:**
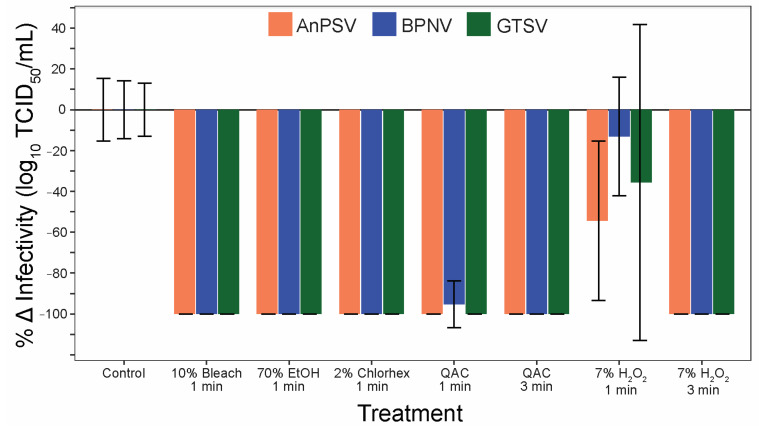
Serpentoviral sensitivity to common disinfectants. The efficacy of five commonly used disinfectants/sanitizing agents against serpentoviruses was assessed by median tissue culture infectious dose (TCID_50_) assay. Standard error bars from average triplicate TCID_50_ measurements are shown.

**Table 1 microorganisms-11-01371-t001:** Cell lines used for serpentovirus in vitro host range experiments.

Grouping	Common Name	Species	Origin Tissue	Commercial Name
Snake-	Diamond Python	*Morelia spilota spilota*	Heart	
Boid	Carpet Python	*Morelia spilota* ssp.	Heart, Kidney, Lung	
	Burmese Python	*Python bivittatus*	Heart	
	Amethystine Python	*Simalia amethistina*	Spleen	
	Common Boa	*Boa imperator*	Heart, Kidney, Lung	
Snake-	Yellowtail Cribo	*Drymarchon corais*	Spleen	
Colubrid	Smooth Greensnake	*Opheodrys vernalis*	Lung	
	Trans-Pecos Ratsnake	*Bogertophis subocularis*	Kidney	
	Corn Snake	*Pantherophis guttatus*	Mass	
Snake-	Venezuelan Rattlesnake	*Crotalus durissus*	Heart	
Viperid	Pygmy Rattlesnake	*Sistrurus miliarius*	Kidney	
	Russell’s Viper	*Daboia russelii*	Heart	VH2
Lizard	Brown Anole	*Anolis sagrei*	Kidney	
	Knight Anole	*Anolis equestris*	Kidney	
	Leopard Gecko	*Eublepharis macularius*	Spleen	
Chelonian	Gopher Tortoise	*Gopherus polyphemus*	Spleen	
	Galapagos Tortoise	*Chelonoidis nigra*	Liver	
	Diamondback Terrapin	*Malaclemys terrapin*	Lung	
	Yellowbelly Slider	*Trachemys scripta*	Heart	
Crocodilian	Chinese Alligator	*Alligator sinensis*	Lung	
	Saltwater Crocodile	*Crocodylus porosus*	Lung	
	Broad Snouted Caiman	*Caiman latirostrus*	Pancreas	
Mammalian	Green Monkey	*Chlorocebus sabaeus*	Kidney	VERO E6
	Feline	*Felis catus*	Kidney	CRFK
	Canine	*Canis lupus*	Kidney	MDCK
Avian	Japanese Quail	*Coturnix japonica*	Fibrosarcoma	QT-35
	Budgie	*Melopsittacus undulatus*	Lipomas	
	Chicken	*Gallus gallus domesticus*	Carcinoma	LMH

**Table 2 microorganisms-11-01371-t002:** Quantification of serpentovirus-infected diamond python heart (DPHt) cells at 32 and 35 °C 48 hpi by epifluorescent immunostaining. Cells exposed to and incubated with the virus were immunostained with an anti-serpentoviral nucleocapsid protein and secondary fluorescently tagged antibody and DAPI nuclear staining. Infected cells and cell nuclei were counted from three representative photomicrographs taken for each condition at 400× magnification.

Virus	32 °C (Infected/To.)	35 °C (Infected/To.)	*p* Value
Mock	0% (0/1322)	0% (0/513)	n/a
AnPSV	27.9% (266/953)	0.5% (4/805)	<2.2 × 10^−16^ *
BPNV	11.6% (122/1053)	0.5% (2/422)	<2.2 × 10^−16^ *
GTSV	15.6% (64/410)	0.4% (3/694)	<2.2 × 10^−16^ *

* Statistically significant via Chi-Square test.

**Table 3 microorganisms-11-01371-t003:** Results from a CPE reduction assay was used to evaluate the effect of ribavirin, remdesivir, favipiravir, NITD-008, enviroximine, pirodavir, and infergen against AnPSV, BPNV, and GTSV.

Compound	Virus	EC_50_ ^a^ (µg/mL)	EC_90_ ^b^ (µg/mL)	CC_50_ ^c^ (µg/mL)	SI_50_ ^d^ (µg/mL)	SI_90_ ^e^ (µg/mL)
Ribavirin	AnPSV ^f^	2.0 ± 0.3	12	>1000	>500	>83
	BPNV ^g^	0.9 ± 0.2	1.6	>1000	>1100	>625
	GTSV ^h^	7.9 ± 3.0	14	>1000	>127	>71
Remdesivir	AnPSV	<0.022 ± 0.02	<0.001	2.2 ± 1.6	>100	>1700
	BPNV	<0.022 ± 0.02	<0.001	2.3 ± 1.8	>105	>1500
	GTSV	<0.022 ± 0.02	<0.001	2.2 ± 1.2	>100	>1700
NITD-008	AnPSV	0.6 ± 0.3	0.76	33 ± 9.0	55	53
	BPNV	0.7 ± 0.4	1.5	37 ± 26	53	13
	GTSV	1.4 ± 0.8	4.8	55 ± 42	39	6.7
Favipiravir	AnPSV	>320	ND	>320	0	ND
	BPNV	>320	ND	>320	0	ND
	GTSV	>320	ND	>320	0	ND
Enviroximine	AnPSV	>0.24	ND	0.24	0	ND
	BPNV	>0.38	ND	0.38	0	ND
	GTSV	>4.9	ND	4.9	0	ND
Pirodavir	AnPSV	1.1	ND	7.3	6.6	ND
	BPNV	2.6	ND	>10	>3.8	ND
	GTSV	0.86	ND	>10	>12	ND
Infergen	AnPSV	>0.01	ND	0.01	0	ND
	BPNV	>0.01	ND	0.01	0	ND
	GTSV	>0.01	ND	0.01	0	ND

^a^ 50% effective concentration (average ± SD, determined by CPE reduction assay), ^b^ 90% effective concentration (determined by virus yield reduction assay), ^c^ 50% cytotoxic concentration, ^d^ 50% selective index (CC_50_/EC_50_), ^e^ 90% selective index (CC_50_/EC_90_), ^f^ Antaresian python serpentovirus, ^g^ Ball python nidovirus, ^h^ Green tree python serpentovirus.

## Data Availability

Data for TCID_50_ analysis is available at https://doi.org/10.6084/m9.figshare.22439266.

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
