# Peer review of "In Vitro Characterization and Antiviral Susceptibility of Ophidian Serpentoviruses"

_microorganisms, 2023, doi:10.3390/microorganisms11061371_

Round 1

Reviewer 1 Report

In the manuscript “In vitro characterization and antiviral susceptibility of ophidian serpentoviruses” Tillis and colleagues isolated and characterized three novel serpentoviruses. The authors leveraged various snake cell lines they have previously developed to both grow and assess the physical and virological characteristics of these agents. Given the dearth of such reagents, the work presented should be lauded for its demonstration of these technologies and their applications. Further, the breadth of the characterization has enormous value for veterinary practice and reptile husbandry. However, there are several points that require elaboration and clarification, and potentially additional experiments that are needed to address the questions that the authors are striving to answer. A list of considerations for the authors is offered below.

Considerations

11. Throughout the manuscript the authors the phrase “limited environmental stability” in reference to the ability of these viruses to retain infectivity for between 10-12 days at room temperature. I would hesitate to ascribe an adjective to this capacity. The virus is stable at room temperature. Unless the authors choose to draw comparison to some agent that is more or less stable, the adjective is unnecessary at best, and undersells a critical aspect of their findings at worst.

 I 2. Line 59: The authors note that the greatest diversity in serpentoviruses is documented in captive python species. There are two points I feel are worth following up here

a.       Is this level of documentation an artifact of sheer numbers (e.g. the relative popularity of pythons as pets which in turn drives breeding) and access (captive vs wild)? If so, I do think this should be noted in the text

b.       While the authors have chosen to focus on the veterinary aspects of these agents, is there any known pertinent ecology that should be addressed? For example; are this agents transmitted transovarially, or are snakes exposed by other snakes during mating or foraging? While this is clearly not the focus of the manuscript, a brief mention of this sort of information would be valuable.

  3. Materials and Methods (Lines 133, 142, and 151): The authors note that DPHt cells were inoculated with virus lysate in these lines while noting elsewhere in the methods that inoculations were performed with “virus”. Is lysate used throughout for all viral inocula? If so, can the authors please provide additional information on the preparation of viral stocks? Is crude lysate used, or is cellular debris filtered?

  4.  Materials and Methods (Single and Multi Step Growth Kinetics): What is the rationale for conducting multiple freeze-thaw cycles prior to titration?

  5.  One-Step Growth Kinetics (Temperature-Dependent Growth Kinetics): One-Step Growth Kinetics are used to measure burst size. Mechanistically, a high MOI is used so that every cell in a dish/plate/flask is simultaneously infected at a given “peak” time point, number of progeny virions are divided by the number of cells at start to determine the number of progeny per infected cell. To address the question that is being asked by the authors on temperature and replication, One-Step kinetics are not appropriate; a Multi-Step growth kinetics experiment is required.

  6. Table 3: There are no units provided for any of the numbers in the table

  7.  Lines 584-598: The authors note that the in vitro finding that the serpentoviruses could infect mammalian cells was unexpected, and note that zoonotic transmission of serpentoviruses or reptarenaviruses have not been reported. It comes across as heavy handed to conclude infection of humans or other mammals by reptile viruses doesn’t take place. It is equally likely that these agents are utterly asymptomatic in mammals but can still infect cells. To substantiate this conclusion serological data documenting absence of exposure via serology in an at-risk population such as veterinarians or those who work in reptile husbandry would be required.

N/A
